# Unifying Categorical Models by Explicit Disentanglement of the Labels' Generative Factors

## Abstract

In most machine learning tasks, the datasets are mainly annotated by categorical labels. For example, in emotion recognition, most datasets rely only on categorical labels, such as "happy" and "sad". Usually, different datasets use different labelling systems (e.g., different number of categories and different names), even when describing the same data attributes. As a consequence, only a small subset of all the available datasets can be used for any supervised learning task, since the labelling systems used in the training data are not compatible with each other. In this paper, we propose a *multi-type continuous disentanglement variational autoencoder* to address this problem by identifying and disentangling the true dimensional generative factors that determine each categorical label. By doing so, it is possible to merge multiple datasets based on different categorical models by projecting the data points into a unified latent space. The experiments performed on synthetic datasets show a perfect correlation between the disentangled latent values and the true generative factors. Also, by observing the displacement of each label's explicit distributions, we noticed that the encoded space is a simple affine transformation of the generative factors' space. As the latent structure can be autonomously learnt by the model, and each label can be explicitly decomposed into its generative factors, this framework is very promising for further exploring explainability in new and existing neural networks architectures.

## 1 Introduction

Supervised machine learning is the most widely used form of machine learning. The possibility to exploit expert knowledge through human annotations allows for simple and direct learning of the data features. As a consequence, the performance and quality of the results obtained by a model are directly linked to the amount of high-quality data available. A clear example of this phenomenon can be seen in machine translation: state-of-the-art models are considered almost indistinguishable from human-performed translation (Popel et al., 2020). However, these results only apply in scenarios in which both source and target languages have a large text corpus available; with low-resource languages, neural methods are far from reaching the same level of performance (Artetxe et al., 2018). In particular, it has been observed that neural machine translation methods have a particularly steep learning curve with respect to the amount of data, obtaining very high-quality results with abundant data, but performing more poorly than traditional statistical methods in low-resource settings (Koehn & Knowles, 2017). Another field, Emotion Recognition in Conversation (ERC), also suffers from the lack of sufficient availability of high-quality data. In Poria et al. (2019) the main challenges of producing ERC datasets are reported: as emotions are subjective topics, it is hard to define an agreed-upon standard on how to generate broadly useful labelled data. The existing models, used to annotate datasets, can be classified according to the following dichotomy:

- **Categorical models:** Each data point is given one or more labels, chosen from a fixed discrete set of possibilities. Therefore, each label defines a cluster of items that share one (or more) common features. As the annotation process is generally performed by humans, the boundaries of each cluster are defined qualitatively and not quantitatively.

Table 1: Emotion taxonomies used by different datasets.

|  | EmoContext | IEMOCAP | Emotionlines | DailyDialog |
|---|---|---|---|---|
| **Neutral** |  | x | x | x |
| **Happiness** | x | x | x | x |
| **Sadness** | x | x | x | x |
| **Anger** | x | x | x | x |
| **Frustrated** |  | x |  |  |
| **Excited** |  | x |  |  |
| **Disgust** |  |  | x | x |
| **Fear** |  |  | x | x |
| **Surprise** |  |  | x | x |
| **Other** | x |  |  |  |

- **Dimensional models:** Each data point is labelled by projecting it into a multi-dimensional continuous space. As a consequence, the annotation is more fine-grained, and it conveys information about the structure of the space itself. Therefore, it is possible to rely on the properties and metrics of a vector space to gain more insights on the data. This is not directly feasible with categorical labels.

Thus, dimensional datasets are more versatile, but more difficult to produce. In fact, it is challenging to guarantee a high inter-annotator agreement during the labelling phase as the model complexity increases (e.g., it is easier to label a color as "orange", compared to assigning to it the three RGB values $(255, 165, 0)$), especially in subjective tasks. For this reason, categorical models are generally preferred over dimensional ones. However, categorical datasets are difficult to work with, as comparison between models is not immediate, since there is not an explicit relation between multiple labelling systems. Therefore, given a specific task, it is not trivial to merge different data sources to obtain a larger shared training corpus. Although the majority of ERC datasets (e.g., IEMOCAP (Busso et al., 2008), Emotionlines (Chen et al., 2019), DailyDialog (Li et al., 2017), and EmoContext (Chatterjee et al., 2019)) consist of utterances and categorical emotional labels associated with them, they are based on different taxonomies. EmoContext, for example, subdivides emotions into four categories, and IEMOCAP uses six of them, while Emotionlines and DailyDialog rely on the scheme proposed by Ekman (1993) (Table 1). On the other hand, there exist few datasets based on the dimensional model proposed by Russell & Mehrabian (1977), according to which each emotion can be described by three independent real values: Valence, Arousal, and Dominance (VAD). An example is the EmoBank dataset (Buechel & Hahn, 2017).

The method proposed in this paper is able to relate categorical models across datasets. As done by Sha & Lukasiewicz (2021), when talking about categorical models, we will refer to each label class (e.g., object, color, ...) as *style type*, and to each possible value within a label class (e.g., dog, cat, red, yellow, ...) as *style value*. We assume that each data point $x$ is generated from the true world simulator Sim given the vector $v = (v_1, v_2, \ldots, v_d) \in \mathbb{V} \subseteq \mathbb{R}^d$ of true continuous generative factors, that is, $x = \text{Sim}(v)$. Therefore, each style type $s$ is determined by a subset $\mathbb{V}_t = \{v_{i_1}, v_{i_2}, \ldots, v_{i_k}\}$ of the generative factors. In addition, we assume that the distributions of each style value representation can be approximated by Gaussian distributions. The architecture performs explicit multi-type disentanglement, preserving the original latent structure of the generative factors. That is, given a labelled sample $(x, t_s)$, where $t_s$ is the style value assigned to the style type $s$, we are able to correctly approximate $p(\hat{v}_{t_s}|x, t_s)$, where $\hat{v}_{t_s}$ is the style representation disentangled from $x$, which lies in a space that is an affine transformation of the original $\mathbb{V}$. Thus, the representation $\hat{v}_{s_t}$, which is obtained by sampling from the obtained distributions or by using a maximum likelihood approach, can be used to infer the true values that determined the feature of $x$ labelled with $s_t$. Being based on a multi-type disentanglement architecture, the method is simultaneously applicable to multiple style types. Therefore, as we are able to compute the true generative factors determining each style value $l$, it is possible to project each data point in a shared dimensional continuous space representing each style type, independently of the used categorical model, thus merging different datasets.

The model is based on an extended version of the architecture proposed by Sha & Lukasiewicz (2021) in which an explicit sampling layer is inserted between the encoder and the decoder of a

standard variational autoencoder model (Kingma & Welling, 2014). The sampling layer learns and stores information related to the ground-truth distributions of the style values for each style type. The encoder generates a content vector and multiple style vectors (one for each style type) that are forced to be close to the correct explicit distributions. Then, for each style type, the sampling layer produces a new style vector that is used by the decoder to reconstruct the original input sample. Compared to the original architecture, instead of relying on a single distribution, we sample a new style vector from each style value's distribution and combine them through a weighted sum, based on a probability score. By doing so, we are able to enforce a smooth transition among similar style values, as each style vector is generated not only by its own style value's ground-truth distribution, but also from neighbour ones. As a consequence, the underlying latent space is continuous.

We conducted experiments on two datasets, dSprites (Matthey et al., 2017) and Base Face Model 2019 (used to generate a collection of almost one hundred thousands sample images) (Gerig et al., 2018), by labelling the data points with different categorical models and trying to reconstruct, through disentanglement, the original real-value generative factors. The results show perfect correlation between the encoded and original values. Furthermore, as each style value's distribution is explicitly stored within the model, we observe that the latent space produced by the model is an affine transformation of the original one. The contributions of this paper are briefly as follows:

- We propose a model that is able to perform an explicit disentanglement, relying exclusively on categorical annotated data. The disentangled style representations lie in a space that is an affine transformation of the true generative factors' one.
- As a consequence, we propose a framework that is able to discover the true generative factors influencing each style type and determining each style value, making it possible to merge multiple datasets that are annotated with different categorical models.

The rest of this paper is organised as follows. Section 2 discusses similar methods in the literature that led to the development of the novel architecture in this paper, highlighting the limitations that it is able to overcome. Section 3 describes in detail the new model. We then present the experiments performed and the obtained results in Section 4 and our conclusive thoughts in Section 5.

## 2 RELATED WORK

Disentanglement is a wide-spread technique, used to isolate the true factors generating the input data, as described in the original $\beta$-VAE paper (Higgins et al., 2017). However, most techniques rely on implicit methods that do not enable for an easy interpretation of the disentangled values. Some components may be pruned (Stühmer et al., 2020) and quantitative measures of disentanglement are still in their infancy (Chen et al., 2018). Latent traversals can provide a qualitative summary of each latent unit's encoded features, but they rely on human expertise and subjectivity and, therefore, may be difficult to apply in real-world scenarios to establish the quality of the obtained disentanglement. In comparison, explicit disentanglement (i.e., the model targets specific factors to disentangle, guided by annotated data) produces easily interpretable components, as it only requires to check that the latent units define the expected features. Since the annotated features are always categorical labels, previous explicit disentanglement works (John et al., 2019; Romanov et al., 2019; Sha & Lukasiewicz, 2021) usually represent the expected feature with a vector, and encouraged it to belong to either of the style values. However, these methods are "uncontinuous", namely, intermediate style values are not encouraged in previous works due to the limitation of categorical labels.

In the emotion recognition task in the conversation domain, Wu et al. (2019) introduced a semi-supervised model (SRV-SLTSM) in order to obtain explicitly disentangled values for the VAD components of each utterance's emotion. However, the architecture requires a percentage of dimensional annotated data to force each latent unit to encode the desired factor, and it relies on the a-priori assumptions of the VAD model. Furthermore, it cannot incorporate the information existing in categorical datasets, as the training process is unsupervised (i.e., the model is an autoencoder and relies only on the input samples $x$ and the model's output to compute its loss). Park et al. (2019) proposed a method to extract the continuous values among the VAD dimensions using exclusively categorical annotations. Their solution outperforms SRV-SLTSM in several metrics, showing the importance of the extra data incorporated into the model, as theorized. Being based on the Earth Mover's Distance

(EMD) loss (Hou et al., 2016), the architecture relies even more on the assumptions made on the latent space and the displacement of the categorical values' distributions in it. In fact, it requires the labels to be ordered and displaced correctly along each axis, according to their true dimensional value (e.g., *joy, sad, happy, anger* would be sorted as (*anger, sad, joy, happy*) on the valence axis). The values used in their experiments are estimated by using the NRC-VAD-Lexicons (Mohammad, 2018) and are fixed throughout the learning process. Our proposed method, instead, is able to obtain an explicit and continuous disentanglement, by relying exclusively on categorical data.

## 3 APPROACH

We will now analyze a single-style type scenario (e.g., where the input data is annotated with a single label); however, the described methods can be applied independently to every style vector. As mentioned above, our approach consists of a modification of the original architecture proposed by Sha & Lukasiewicz (2021), which predefined a Gaussian distribution for each style type value, and, given a sample $(\boldsymbol{x}, t)$, required every disentangled $d$-dimensional style representation $\boldsymbol{s}$ generated by the encoder to belong to the style distribution that represents the annotated style type value $t$. This is achieved by maximizing the probability

$$p(t|\boldsymbol{s}) = \frac{p_{Nor}(\boldsymbol{s}|t)p(t)}{p(\boldsymbol{s})}, \tag{1}$$

where $p(t)$ is the prior distribution of the style values, and $p_{Nor}$ is as follows:

$$p_{Nor}(\boldsymbol{s}|t) = \frac{\exp(-\frac{1}{2}(\boldsymbol{s} - \mu_t)^{\mathrm{T}}\Sigma_t^{-1}(\boldsymbol{s} - \mu_t))}{\sqrt{(2\pi)^d \det(\Sigma_t)}}. \tag{2}$$

Then, a new style vector $\boldsymbol{s}'$ is sampled from its corresponding correct distribution, according to the input label, and forwarded to the decoder. To make sure that each style distribution actually encodes the correct style representation, they sample from their corresponding distributions a vector $\boldsymbol{s}'_j$ for each $j \in 1, \ldots, m$, being $T = \{t_1, t_2, \ldots, t_m\}$ the set of possible style values. Then, each $\boldsymbol{s}'_j$, and thus its distribution, is forced to be similar to the correct style value representation through the following classification loss (to be computed for each style type):

$$L_{\mathrm{CL}} = -\frac{1}{|T|}\sum_{j=1}^{|T|}\log(p_c(t_j|\boldsymbol{s}'_j)), \tag{3}$$

where $p_c$ is defined as:

$$p_c(t_j|\boldsymbol{s}'_j) = \frac{p_{Nor}(\boldsymbol{s}'_j|t_j)p(t_j)}{\sum_{t' \in T} p_{Nor}(\boldsymbol{s}'_j|t')p(t')} = \frac{p_{Nor}(\boldsymbol{s}'_j|t_j)}{\sum_{t' \in T} p_{Nor}(\boldsymbol{s}'_j|t')}. \tag{4}$$

since we assume that all labels share the same prior probability. The vector $\boldsymbol{s}'$, however, cannot contain any of the unique nuances of the input data point that could be represented in $\boldsymbol{s}$: as it is obtained from a single style value's distribution, it can only encode the average features shared by all the samples with the same style value.

### 3.1 ARCHITECTURE

The general structure of our architecture is unchanged compared to the one employed by Sha & Lukasiewicz (2021): it consists of an encoder, a decoder, and a bottleneck layer, in which a resampling operation happens for each of the generated style vectors. In order to obtain a continuous disentangled representation, we modified the sampling layer. Consider a sample $(\boldsymbol{x}, t)$ and its generated embedding $\boldsymbol{h}$, split into content vector $\boldsymbol{c}$ and a single style vector $\boldsymbol{s}$, whose possible $m$ style values are the elements of $T$. As described above, a probability distribution $\mathcal{N}_i$ is associated with each of them. Sampling a new $\boldsymbol{s}'$ from a single distribution discards any extra information (i.e., detailed features other than the discrete style category) stored in $\boldsymbol{s}$. Our method preserves it by sampling a different style vector $\hat{\boldsymbol{s}}_i$ from each distribution $\mathcal{N}_i$ and summing them to $\boldsymbol{s}'$, weighted by each probability distribution $p(t_i|\boldsymbol{s})$. Formally, the new resampled vector $\hat{\boldsymbol{s}}'$ is obtained as follows:

$$\hat{\boldsymbol{s}}' = \boldsymbol{s}' + \sum_{i=1}^{j}\hat{\boldsymbol{s}}_i\,p(t_i|\boldsymbol{s}). \tag{5}$$

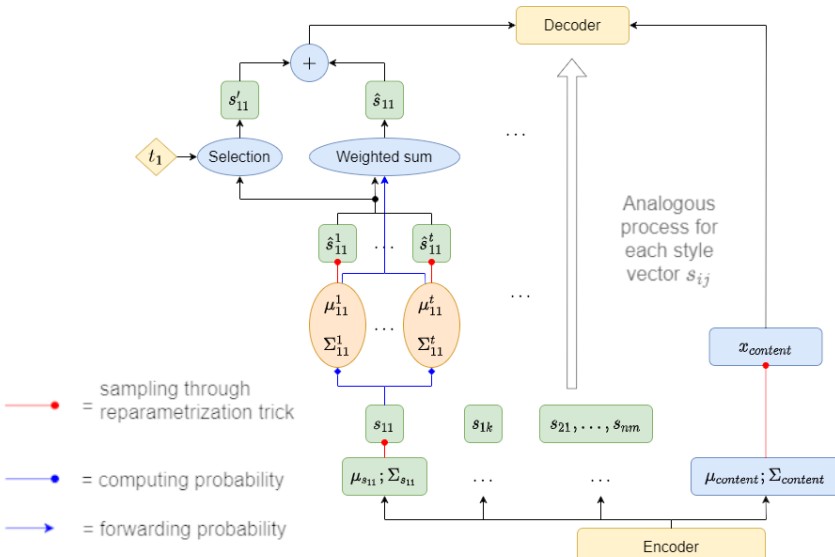

Figure 1: Architecture of our method: the ellipses represent the $\mu$ and $\Sigma$ of the style value distributions used to sample the new style vectors.

The vector $\hat{s}'$ is then fed to the decoder. By doing so, we obtain a continuous latent space, as each distribution contributes to the final representation, and therefore similar style value encodings are forced to be close also in the latent space. In addition, we can encode intermediate style features by interpolating the representations obtained from different distributions, since $\hat{s}'$ is a weighted sum of them. In order to disentangle the true independent generative factors instead of an alternative representation of the style values, we used one-dimensional style vectors. This also defines a clear notion of order, induced by the real line. As it is possible for a style value to be multidimensional (i.e., being determined by the combination of multiple generative factors), we allow for multiple style vectors to represent the same style value, each learning to encode a different generative factor. As a consequence, we have to modify the classification loss proposed by Sha & Lukasiewicz (2021) as follows (remembering that we are considering a single style type):

$$\hat{L}_{\text{CL}} = -\frac{1}{|T|} \sum_{j=1}^{T} \log(\hat{p}_c(t_j | \boldsymbol{s}'_{j1}, \ldots, \boldsymbol{s}'_{jk})), \qquad (6)$$

where $k$ is the numbered of style vectors reserved to encode a style value, and $\hat{p}_c$ is as follows:

$$\hat{p}_c(t_j | \boldsymbol{s}'_{j1}, \ldots, \boldsymbol{s}'_{jk}) = \frac{\prod_{i=1}^{k} p_{Nor}(\boldsymbol{s}'_{ji} | t_j)}{\sum_{t' \in T} (\prod_{i=1}^{k} p_{Nor}(\boldsymbol{s}'_{ji} | t'))}. \qquad (7)$$

Furthermore, assuming a $d$-dimensional style type, with $d < k$, our method is able to find $d$, reporting $k - d$ style vectors as *unused*. By measuring the KL divergence between a style type's distributions, it is possible to quantify the amount of information stored in each latent unit. When the KL divergence is close to $0$, there is no difference between the style representations among that dimension, and the resampled value encodes only random noise and remains unused. This allows us to better understand the structure of the generative factors' hidden space. The complete diagram of the architecture is given in Figure 1.

## 3.2 DATASETS

As we focused on defining a general-purpose framework for disentanglement and explainability, we tested our method on synthetic datasets used in disentanglement-related tasks. In order to obtain experimental results, we created a modified version of the dSprites (Matthey et al., 2017) and Base Face Model 2019[1] (Gerig et al., 2018) datasets. In both cases, we replaced some of the continuous

---

[1]To construct the dataset, a collection of 96100 data points was generated, using 100 different face IDs and 31 evenly distributed values for both the face orientation ("yaw-range") and illumination angle ("illu-range").

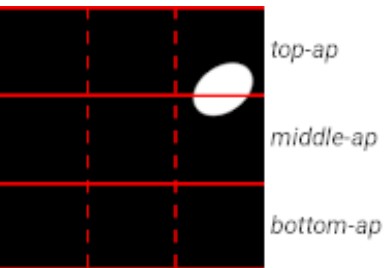 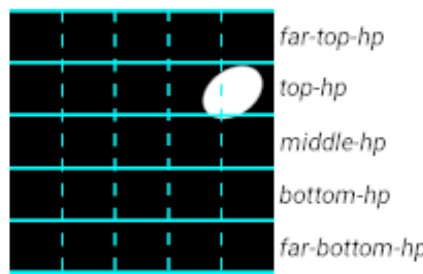

Figure 2: Visualization of the dSprites_AP (left) and dSprites_HP (right) categorical systems on the $x$ (solid color) and $y$ (dotted line) coordinates. The labels refer to the $y$ value. Given the image data *img*, the model receives as input the tuple (*img*, *right-ap*, *top-ap*) or the tuple (*img*, *far-right-ap*, *top-hp*), depending on the categorical model chosen.

values used to generate each data point with specific categorical labels, based on predefined ranges among each dimension. The model is then given access only on the sample image and its categorical labels, while the true generative values are used as ground truth to be compared with the obtained results in order to evaluate the model's performance. dSprites is a dataset composed by 737280 square images of $64 \times 64$ pixels. Each image is determined by five factors that can assume different values in a specific range: *shape, scale, rotation, $x$, and $y$*. We defined two categorical models. Each uses two one-dimensional style types (i.e., that rely on a single generative factor) in order to label the $x$ and $y$ values separately. A third one, instead, defines a single two-dimensional style type to describe the combination of the $x$ and $y$ coordinates. The ranges and labels used for each categorical model are as follows:

- **dSprites_AP** (Average Precision): The $x$ value (in pixels) is labelled according to the ranges $R_{AP} = \{[0, 23], [24, 39], [40, 63]\}$ corresponding to the three style value names *left-ap, center-ap,* and *right-ap*. Analogously, $y$ is labelled according to $R_{AP}$, using the style value names *top-ap, middle-ap*, and *bottom-ap*.

- **dSprites_HP** (High Precision): The $x$ value is labelled according to the ranges $R_{HP} = \{[0, 10], [11, 25], [26, 37], [38, 52], [53, 63]\}$ and the style value names *far-left-hp, left-hp, center-hp, right-hp,* and *far-right-hp*. Analogously, $y$ is labelled according to $R_{HP}$, using the style value names *far-top-hp, top-hp, middle-hp, bottom-hp,* and *far-bottom-hp*.

- **dSprites_Co** (Compound): The $x$ and $y$ values are labelled as a single feature according to the ranges $R_{Co} = R_{AP} \times R_{AP}$. This defines nine style value names, defining a $3 \times 3$ grid over the input images. The used names are *ij-co*, each identifying the grid cell at row $i$ and column $j$.

A single shape type was used to generate the dataset associated with each categorical model (e.g., dSprites_AP is built using the elliptic shapes). A visualization of the dSprites_AP and dSprites_HP categorical models is given in Figure 2. The dataset generated using the Base Face Model 2019 was processed in a similar way, defining five evenly sized ranges for the face orientation (i.e., the "yaw-range" value, ranging from $-90$ to $+90$ degrees) and three for the illumination angle (i.e., the "illu-range" value, ranging as well from $-90$ to $+90$ degrees). A total of 96100 images were generated and labelled, defining the BFM_2019 dataset.

## 4 EXPERIMENTS AND RESULTS

To our knowledge, there are no previous works that analyze the relation between explicit disentanglement and categorical annotations. Therefore, we describe the metrics used to evaluate the model, and we briefly introduce the results obtained by using the original architecture by Sha & Lukasiewicz (2021), in order to establish a baseline. As we are interested in the structure of the latent space in which the disentangled style representations are embedded, we focused on the displacement of the distributions and the performance in encoding and reconstructing intermediate style values. As our method relies on one-dimensional style vectors (i.e., scalar variables), we used the same one-dimensional style encoding also for the architecture by Sha & Lukasiewicz (2021) to

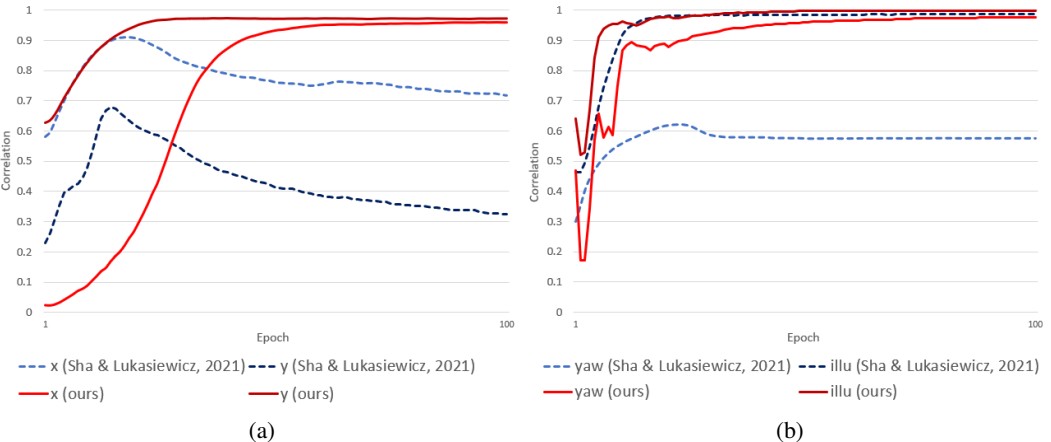

Figure 3: Correlation between the style distribution means and true label range average value for each disentangled style type in the dSprites_HP (a) and BFM_2019 (b) datasets. The "illu" style type has only three possible style values, and therefore it is straightforward for both architectures to achieve optimal results.

Table 2: Ranges encoded by each style value's distribution compared to the true label values (the latent space is clipped according to the distributions' standard deviation, considering one standard deviation away from the mean.)

| Style value | Distribution's range | Size % | Label's range | Size % |
|---|---|---|---|---|
| *far-left-hp* | $(-2.32, -1.45]$ | 19.8% | $[0, 10]$ | 17.2% |
| *left-hp* | $(-1.45, -0.72]$ | 16.6% | $[11, 25]$ | 23.4% |
| *center-hp* | $(-0.72, 0.11]$ | 18.9% | $[26, 37]$ | 18.8% |
| *right-hp* | $(0.11, 0.98]$ | 19.8% | $[38, 52]$ | 23.4% |
| *far-right-hp* | $(0.98, 1.94)$ | 21.8% | $[53, 63]$ | 17.2% |

get a fair comparison. It is true that, by using multidimensional style vectors, better results could be achieved. The encoding latent space, however, would be completely unrelated to the original generative factors, and a comparison between different categorical models unfeasible.

We trained the architecture by Sha & Lukasiewicz (2021) on the dSprites_AP and dSprites_HP datasets and analyzed the style representations in the latent space. By measuring the correlation between each style value distribution's mean and each label's range average value (e.g., *far-left-hp* represents $x$ values in the range $[0, 10]$, so its average value is 5), we observed that the distribution displacements do not reflect the order induced by their true generative factors, resulting in a low correlation. For example, considering dSprites_HP and the $x$ coordinate, we require the ordered sequence (*far-left-hp, left-hp, center-hp, right-hp, far-right-hp*)[2] in order to obtain a continuous encoding space. Our method, instead, achieves an almost perfect correlation for every style value on both datasets (Figure 3), forcing the distributions to reflect the order of the ground-truth labelling sequences. This behaviour is consistent across several runs of the algorithm. In contrast, the results using the architecture by Sha & Lukasiewicz (2021) suffer from a high variance, as any random initialization of the model's distribution parameters leads to a different displacement, and thus correlation. Appendix A shows several examples of the distributions' displacement for both datasets. This is possible, as the parameters defining each distribution are explicitly stored inside the model.

Having a continuous encoding space is necessary to represent intermediate style values. To show this, we performed latent traversals on the embedded vectors produced by the architectures trained on the dSprites_HP dataset. The model by Sha & Lukasiewicz (2021) encodes each style value as an independent group of data points. There is no smooth transition for intermediate representations and the discontinuities in the latent space can be clearly observed (Figure 4a). In comparison, our method shows an ordered sequence of data points on both the $x$ and $y$ traversals (Figure 4b).

---

[2]As we are dealing with linear transformations, also the reversed sequence would be acceptable

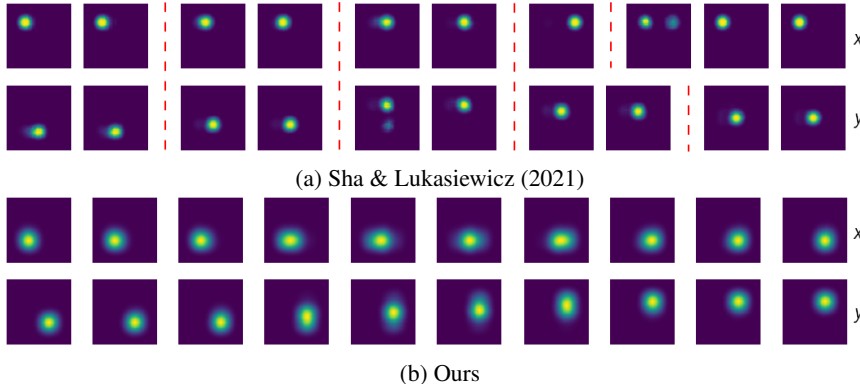

(a) Sha & Lukasiewicz (2021)

(b) Ours

Figure 4: Latent traversals performed on dSprites_HP by fixing all the latent units except the ones encoding the $x$ and $y$ values, whose value is increased from left to right. In (a), the discontinuity points can be clearly seen in the reconstructed samples, which fail to encode intermediate style values, and instead present two shapes. There is no smooth transition between different style values. Instead, it is possible to identify the groups defined by the five style values as each one produces almost identical images (b) shows a smooth transition across both dimensions.

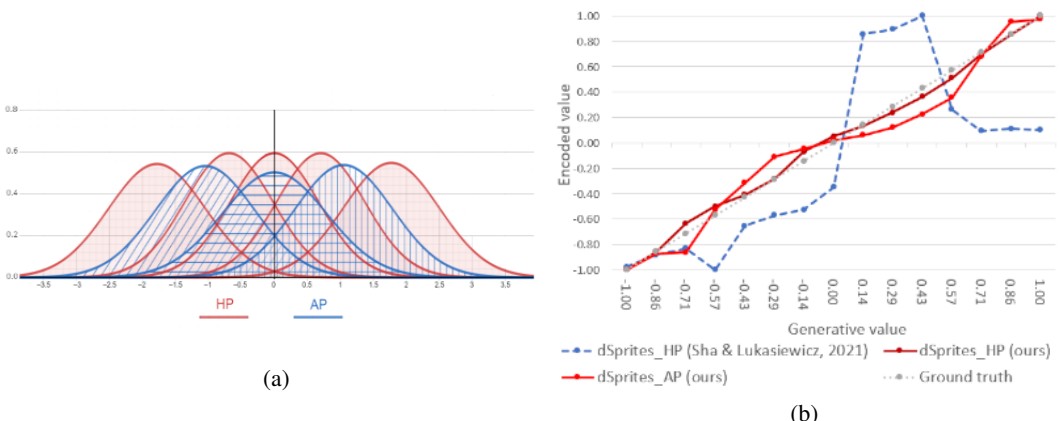

(a)

(b)

Figure 5: (a) Style value representation density functions of the $x$ coordinate for both dSprites_HP and dSprites_AP. The distributions are displaced in the correct order (i.e., *far-left-hp, left-hp, center-hp, right-hp, far-right-hp* and *left-ap, center-ap, right-ap*) and overlap nicely to represent intermediate values. (b) Values encoded in the latent unit corresponding to the $x$ style value. Our method shows a high correlation with the ground truth, across both datasets.

In addition, we compared the relative size of the ranges used to define each style value with the ranges defined by the model, obtained by considering the maximum likelihood style distribution for each possible value of the latent space. Table 2 reports the results obtained by our method on the $x$ coordinate in the dSprites_HP dataset. There is a high correlation between the two spaces, showing that the encoded latent space is an approximation of an affine transformation of the original one. As a consequence, it is possible to train a single model on both the dSprites_AP and dSprites_HP datasets simultaneously, by using two sampling layers (one for each dataset) that share the same encoder and decoder. The result is that the distributions in both layers belong to the same space. The representations of similar style values in both categorical models are, therefore, similar even in the latent space (Figure 5a). This would not happen if each distribution was learnt independently, since it would be displaced randomly and not uniformly across datasets. If we compare the normalized latent values for the $x$ generative factor, we see that our method produces similar encoding for both dSprites datasets, highly correlated with the ground-truth $x$ value. This does not happen using the architecture by Sha & Lukasiewicz (2021), as each style value representation is independent and does not follow a precise structure (Figure 5b).

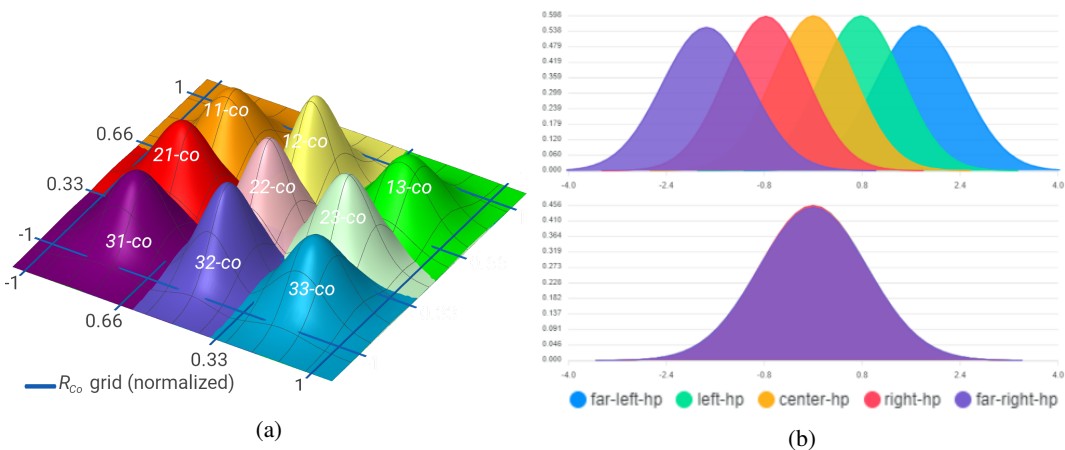

(a)            (b)

Figure 6: Displacement of the style distribution in the latent space. (a) shows the disentanglement of the generative factors among each dimension of a two-dimensional style type. (b), instead, demonstrates how, when $k > d$, the distributions in the unused dimensions do not encode any information, as they are completely overlapping, and therefore every style representation appears as the same.

We used the dSprites_Co dataset to study the disentanglement of multi-dimensional style types. Since the labels used in the dataset are determined by the combination of the two independent factors $x$ and $y$, we defined the style type "position" with nine possible style values (i.e., the labels $ij$-$co$, $i \in \{1, 2, 3\}$, and $j \in \{1, 2, 3\}$). We set $k = 2$ and expected the model to disentangle the two generative factors on the two available dimensions. By normalizing each dimension and comparing the displacement of the distribution with the grid induced by $R_{Co}$, we observe a high correlation with the ground-truth values (Figure 6a), confirming the capability of the model to disentangle them. In addition, we analyzed the scenario in which $k > d$ to show how our method uses only the strictly necessary number of dimensions to represent a style value, corresponding to the underlying number of generative factors. By setting $k = 2$ and training on dSprites_HP, we observe that the encoding for the style type factor $x$ (and analogously for $y$) only uses one of the two available dimensions, as expected. In fact, the distributions in the second dimension associated with the style type completely overlap with each other, having a pairwise KL divergence approximately equal to $0$ (Figure 6b). As the model has to sample a new style vector from its corresponding style distribution, and given that all the probability densities are equal, we conclude that the latent unit can only encode random noise, not being able to distinguish different style values.

## 5 CONCLUSION

We have shown that current methods performing explicit disentanglement are not suitable to identify the true generative factors corresponding to each style type. Instead, an independent representation in a different latent space is learnt. Since there is no enforced structure on the new space, the style representations depend exclusively on the random initialization of the model parameters. As a consequence, it is not possible to project different categorically annotated datasets into the same dimensional space. Furthermore, as the focus of the architecture by Sha & Lukasiewicz (2021) is on each single style value, seen as a category of objects that share the same feature, the model fails to represent the unique characteristics of each input sample, and therefore to provide a smooth transition between similar style values. To fix these disadvantages, we obtain a unified style representation across different categorical models, and making it consistent with the generating space by forcing an ordering among the distribution on each encoding dimension. In addition, since similar style values are now also close in the latent space, our model is able to encode individual nuances of each data point. By doing so, it is possible to label each data point with its true dimensional attributes, allowing for multiple categorical datasets to be merged together.

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

## A APPENDIX

We report the displacement of the distribution in the latent space for both architectures and the different datasets (Figure 7 and 8).

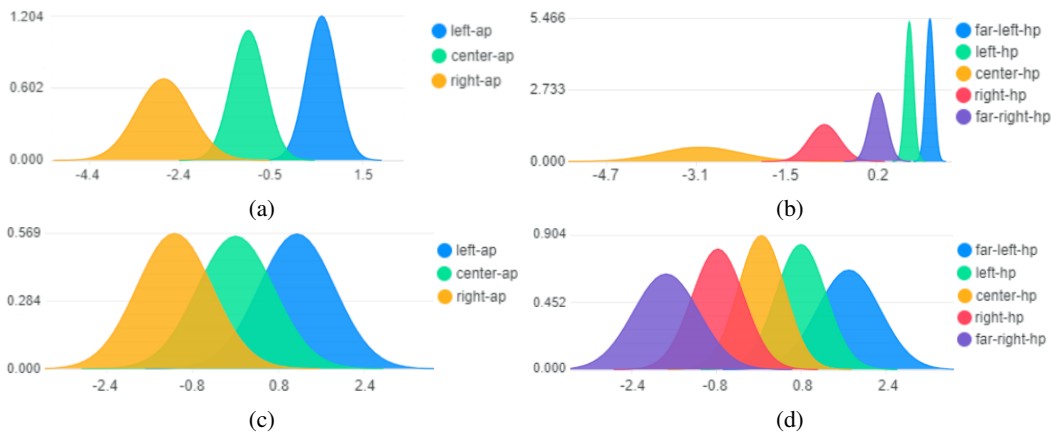

Figure 7: Explicit distributions of the style representations along the $x$ coordinate in the (a) dSPrites_AP and (b) dSprites_HP datasets, using the architecture developed by Sha & Lukasiewicz (2021). Analogously, (c) and (d) report the distributions' displacement achieved by our new method on the same datasets. The charts are constructed from the $\mu$ and $\sigma^2$ values stored in sampling layers of the models. Similar results are obtained along the $y$ coordinate.

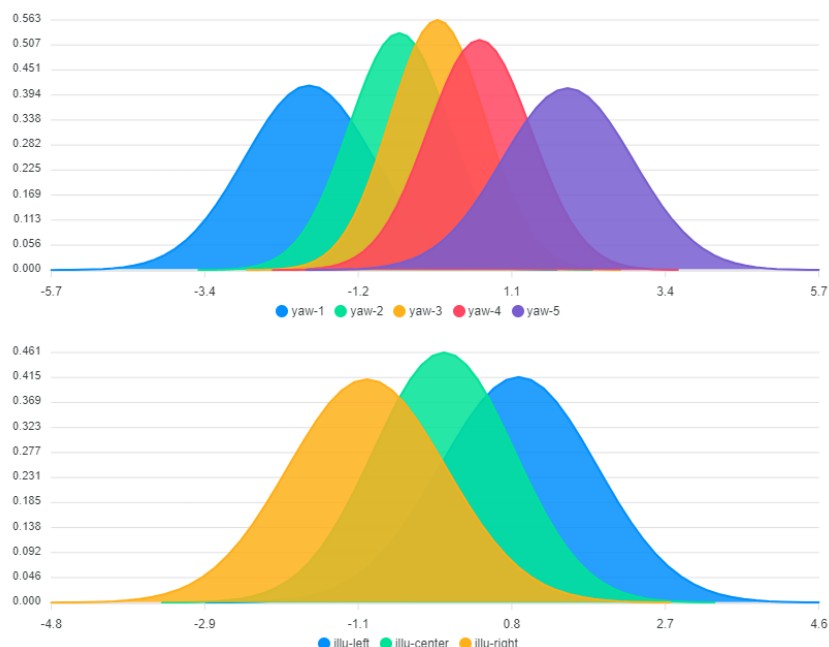

Figure 8: Explicit distributions of the style representations for the *yaw* and *illu* style types in the BFM_2019 dataset. The displayed order correctly reflects the ground-truth generative values (up to an affine transformation).

