# OpenReview forum: "Unifying Categorical Models by Explicit Disentanglement of the Labels' Generative Factors"
_ICLR.cc/2022/Conference — ICLR 2022 Submitted_

### Official Review · Reviewer_ugco · 2021-10-27

**Correctness:** 3
**Technical Novelty And Significance:** 2
**Empirical Novelty And Significance:** 2
**Recommendation:** 3
**Confidence:** 2

**Main Review:**

Overall the paper seems like a very incremental improvement over SL21. It's a small modification on the overall architecture, and the evaluation is done only on a single synthetic dataset. I was a little confused also about the distinction between a single multidimensional style vector and many scalar style values; the paper seems to argue that multiple scalars provide better disentanglement than a single vector but I couldn't follow what this means in practice in terms of the model equations.

The evaluation is very sparse; it's unclear to me how to interpret these results on small synthetic datasets. The results don't seem substantially better than the SL21 baseline, and it's also not clear how to apply these models to other problems meaningfully.


**Summary Of The Paper:**

The paper is a follow-up to Sha and Lukasiewicz 2021 "Multi-type Disentanglement without Adversarial Training" (SL21 from now on) which uses a slight modification of the technique from SL21 to the new task of training a single model on multiple labeled datasets with different label sets that encode the same information at different levels of granularity (so in a way it's a variant of multi-task learning). The model effectively learns a single continuous embedding for examples from which all label sets are easily predictable, effectively unifying the different label sets. The experiments are performed on modified versions of the dsprites and base face datasets. On dsprits it shows a more smooth representation than SL21.

**Summary Of The Review:**

Overall the paper seems very incremental on top of a single baseline, and the evaluation is very sparse. I don't see the novelty or significance here.

---

> ### Author Response · Authors · 2021-11-16
> **Clarification of the expressed doubts.**
>
> Thank you for your review.
>
> I'll try to address the points you highlighted in your review:
>
> - single vector and multiple scalar values: what we argue is that both multi-dimensional vectors and multiple scalar values provides the same disentanglement capabilities, if we consider the canonical definition of "separating a specific style type from the remaining content related information". However, previous works, by focusing on the whole vector representation, obtained disentangled styles that are unrelated to the true "ground truth" representation. As a consequence each model (and each individual training iteration) creates an independent latent representation, based on the original random initialization of the model (as seen in SL21). This does not allow us to merge different datasets.
> The key idea of focusing on the individual scalar dimensions of each vector allows to impose stricter constraints to obtain a latent representation that is aligned with the original one (up to affine transformations). We can for example speak of the order of the distributions and clearly understand the result of their weighted sum (instead, when considering the whole vector, a possible weighted sum would equally affect all the dimensions at the same time, and therefore limit the expressivity/interpretability of the model).
>
> - In our opinion the results show a clear improvement compared to the original SL21. For example, by looking at figure 5b, we can clearly see that by using SL21, merging datasets would simply not be feasible as the model would output completely arbitrary values for the x value. On the contrary, our model is able to recreate the original x value on both datasets, introducing only a little noise (the small y variation compared to the diagonal ground truth). Furthermore (despite our work not focusing on the reconstruction quality), by looking at figure 4a/b, we can see that our model is able to correctly reconstruct even samples that are intermediate between two style values, by interpolating through them. SL21 is not able to do so (e.g., picture 8 in the first line), as those sample does not clearly belong to any style value, and are considered to be outliers by the model. In general, our results show that our model is able to achieve a better accuracy by representing intermediate style values. This is possible since we aim to reconstruct the ground truth feature latent space, instead of disentangling each style type in a new, independent space. As a consequence, we are also able to infer key properties of the underlying latent space and, therefore, consistently project different categorical models to it (allowing us to merge multiple datasets).
>
> - Our method can easily be applied to any scenario in which we have different kinds of labels representing the same data. By discovering the number of generative factors underlying those labels, it is possible to project them in the same continuous space and therefore merge them into a single datasets (our model can be easily plugged into any training pipeline as a "data augmentation" pre-processing step). As having more data allows to train deeper and more complex models, we hope that our method, being general purpose, could bring benefits to several real world tasks. A direct field it could be applied to, as mentioned in the introduction, is emotion recognition. We aim, in future works, to provide results on downstream tasks using our technique to augment the quantity of training data available. However, in this work, we preferred to focus on the general properties of the proposed method (and therefore use synthetic datasets), as each one of them could be suitable for a different scenario.

---

### Official Review · Reviewer_w1er · 2021-11-02

**Correctness:** 3
**Technical Novelty And Significance:** 2
**Empirical Novelty And Significance:** 2
**Recommendation:** 3
**Confidence:** 4

**Main Review:**

The proposed method is as a minor variation of Sha and Lukasiewicz (2021). Specifically, the main modification is that instead of sampling from only the ground-truth latent distribution for the decoder, a weighted sum is used over all style values. Hence, the technical contribution of this paper is not as high as other papers.

The explanation of the method in Section 3 is not very clear, and it took a couple of reads to understand the method. For instance, what is the overall loss that the model is trained on? If it is a VAE objective, then it can be stated in the first part of Section 3, and all elements of the losses can be explained one by one. In the current form, the text misses a combination of all mentioned loss factors.
Additionally, it is unclear why the selection sample $s'$ and the weighted sum are summed together to $\hat{s}'$. What does $\hat{s}'$ intuitively represent? For a model with perfect accuracy, $\hat{s}'$ is twice $s'$. Is it just a simple way of combining the selection and weighted sum? Can there be any issues with this setup?
Next, in the introduction, it is multiple times mentioned that the model is able to 'correctly approximate' certain distributions. These claims initially suggest that it can be proven that the model always converges to the correct distributions, but the rest of the paper suggests that this is empirical. Further, the latent distribution $p(v|x,t)$ can probably not be found up to isomorphism classes like translation or scaling, but the paper misses a discussion of such.

Furthermore, the presented experiments do not sufficiently evaluate the proposed method. Firstly, the method is only compared to Sha and Lukasiewicz (2021) on the aspects that the proposed modifications were intended to improve on. Since the method of Sha and Lukasiewicz (2021) was not explicitly proposed for this, it is not unsurprising that the proposed method outperforms the baseline. However, no other was compared to it. While there might be not many works on explicit disentanglement and categorical annotation, the experiments need to convey the reader of the usefulness of the proposed method. This can be done on downstream tasks, where it can be compared to many other works. For instance, the motivation given in the abstract was that different labelling systems across datasets do not allow the joint supervised training on all of those. Nonetheless, one straight-forward way of doing it is to train a model with multiple classification heads, one per dataset. The proposed method can then be compared to this setting to show its benefit of transforming the classification into a continuous space.
The second issue is that the experiments rely on datasets that were created for this paper specifically and not standard benchmarks. While the fundamental data comes from benchmarks, the task setup is not. Similarly, as in the first point, this does not convey any usefulness of the method.
Finally, the introduction's motivation is heavily based on emotion recognition, even presenting a table of such in Table 1. Still, none of the experiments in the paper touch upon this area, which convincingly is an area where categorical annotations are difficult. This leaves the reader confused in terms of why the method has been proposed with this motivation if it might not be able to help in emotion recognition.

__Minor comments__
- Eqn 6 is likely missing a log
- Figures look in several parts blurry, it is better to use vector graphics where possible
- Page 3, second paragraph: typo in 'Basel Face Model'
- A visualization of the second dataset (BFM) is missing, which would help in getting an intuition for the dataset

__Post-rebuttal update__
I appreciate the authors' response and changes to the submitted paper. While I agree that the paper should not just focus on a single downstream task, I believe it is important to have an experiment at least one of these domains so that the method can be compared to more methods (e.g. the simple multi-head training) to clearly show its benefits. The current experimental evaluations are not sufficient to fully convince the reader. Hence, I strongly recommend including such an experiment in a future version, and keeping my initial score.

**Summary Of The Paper:**

This paper proposes a method for projecting datasets with different categorical labels into a unified latent space. It is based on an architecture proposed by Sha and Lukasiewicz (2021) which is a VAE with a structured latent space. The latent space is split into a content and style part, where the style part models the classification of labels, and the content the rest of the features to reconstruct the input. The goal of the proposed modification is to have similar label values close together in latent space while disentangling across labels. In experiments on synthetic datasets, the model is shown to provide the specified properties.

**Summary Of The Review:**

In summary, the minor modification of the architecture proposed in previous work and the limited experimental comparison are not sufficient to pass the acceptance level for the paper. Thus, my current recommendation is 'Reject'.

---

> ### Author Response · Authors · 2021-11-22
> **Clarification of the expressed doubts.**
>
> Thank you for your review and, in particular, the suggestion for further experimental comparison you suggested. We will try to provide more specific benchmarks in future work. We uploaded an updated version where the mistakes you highlighted are fixed.
>
> To clarify your doubts:
> - \hat{s}' represent the weighted sum of different style representations, as a consequence it is possible to represent intermediate style values, which is not possible in the model by Sha&Lukasiewicz, as it can be seen in picture 4. As a consequence, for specific samples it will be different by s', providing more information of the nuances of the single data sample.
> - The loss is based on the complex loss function introduced by Sha&Lukasiewicz, modified when needed as explained, to suit the architectural changes.
> - As mentioned the latent distribution can be found up to an affine transformation of the true one, as a consequence when comparing the results and the encoded value to show it is possible to merge different datasets, we employed a normalized version of the embeddings.
>
> We understand that our comparison is only against another single model but, as you mentioned, there are not many works focusing on this topic. With our work, we hoped to establish a baseline for future comparisons. We believe that the properties of our model can be applied to many problem instances and are not reliant on any assumption of the data structure. As a consequence we preferred to avoid focusing on specific downstream tasks in favour of highlighting more general results (e.g., the possibility of understanding the dimensionality of the latent space underlying a categorical label) that do not affect benchmarks performance directly.

---

### Official Review · Reviewer_RNFG · 2021-11-03

**Correctness:** 4
**Technical Novelty And Significance:** 3
**Empirical Novelty And Significance:** 2
**Recommendation:** 6
**Confidence:** 3

**Main Review:**

Strengths:
 It is interesting to split the multiple factors in variational autoencoder. The proposed method is effective in learning disentangled factors.

Weakness:
It would be better if the authors provide  intuitive explanation of some key notations when presenting details of the methods. For example, what is the intuitive meaning of P_Nor (Eq.2). What are the differences between s, \hat{s}, and s^\prime? Are all of them the factor representations?

Equ 4 is similar to bayes but missing a prior probability p(tj). Please explain why p(tj) is omitted.

It would be better if the authors evaluate the methods on emotional data, as they mentioned in the abstract and introduction.

**Summary Of The Paper:**

The authors consider multiple disentangled factors in a variational autoencoder. The proposed method extends the structure of Sha&Lukasiewcz(2021) by modifying the sampling layer. In the new sampling layer, multiple factor representation is generated as well as a probabilistic weight for each factor. Then, the final representation is a weighted sum of all the factors.

The proposed method is evaluated on toy data, i.e., modified dSprites and Base Face Model 2019. The results show that the method is able to encode multiple factors.

**Summary Of The Review:**

The idea is interesting and reasonable.
However, the experiments on real data is suggested.

---

> ### Author Response · Authors · 2021-11-10
> **Clarification of the expressed doubts.**
>
> Thank you for your review.
>
> Regarding your first point:
> For better understandability, we employed the same notation used by Sha&Lukasiewicz(2021).  P_Nor is the probability, based on each style value normal distribution, that a generated style vector s belongs to style value t. The idea is to maximize the probability of s to belong to the correct style value, while minimizing the other ones. Therefore, by computing P_Nor(t|s) for each t, we can determine which is the most likely style value predicted by the model and compute the associated error. s, \hat{s} and s' all indeed all representation of the style value disentangled by the model. More in detail:
> - s is the value generated by the encoder
> - s' is the value obtained using the method by Sha&Lukasiewicz by sampling a vector from the correct style value distribution. It is also the value that, in the original model, is fed to the decoder
> - \hat{s}' is instead the style representation that we propose that employs a weighted sum of the samples drawn by each style distribution (imagine instead of sampling a single s' from the correct distribution given the label, we sample one of them from each distribution and weight their importance). This is the value that we pass to the decoder
>
> Regarding the second point:
> In eq.4 we omitted p(t) and p(t') in the numerator and denominator as they cancel each other (we assumed that the prior probabilities of the labels are the same, we agree that this is not clear in the paper, and we'll add a mention to it).
>
> Regarding the third point:
> We agree that experiments on real data are necessary. However, we preferred to focus on more abstract properties of the model for the following reason: as the problem analysed (i.e., how to merge categorical and dimensional systems through disentanglement) is not really explored in literature it is difficult find general benchmarks to use as a baseline. Instead, as suggested by other reviewers, the quality of the results could be established using downstream tasks on, for example, emotion recognition. However, this would require establishing less direct performance metrics than the one used in our paper. Considering emotion recognition, for example, the idea is to generate a merged dataset and train current classification methods on it and observe if there's any performance benefit. However, as there is no guarantee that the disentangled representation would be aligned with the VAD model (and therefore the generated continuous labels would be difficult to explain), it would be difficult to explain any gain (or loss) in performance. Studying this phenomenon would shift the focus of the paper onto a specific topic. Instead, by using synthetic datasets we can clear explain and justify our results and provide insights on other model's properties (e.g., discovering the dimensionality of the latent space) that are applicable in different scenarios, not only emotion recognition.

---

### Official Review · Reviewer_4rCs · 2021-11-03

**Correctness:** 4
**Technical Novelty And Significance:** 2
**Empirical Novelty And Significance:** 2
**Recommendation:** 3
**Confidence:** 4

**Main Review:**

[Strengths]
1. The model shows perfect correlation between latent values and true generative factors on the synthetic data.
2. The approach is simple and mathematically sound.

[Weaknesses]
1. The novelty of the approach is very thin. Model is a simple extension of disentangled variational autoencoder using a weighted sum. The idea of using weighted sum instead of single distribution is common in many models including Transformer and GAN. The application to variational autoencoder is straightforward.
2. The motivation in abstract and title states the need to merge multiple datasets. However, the experiments are only on individual synthetic datasets and are limited to disentanglement use case. Showing examples of using the model to merge datasets and getting improved state-of-the-art would strengthen the claims and support motivation.
3. The writing of the paper is unclear in several sections. Understanding the approach requires multiple readings as the intuition is not provided well.

**Summary Of The Paper:**

This work proposes a continuous disentanglement variational autoencoder. The approach is an extension of the disentangled variational autoencoder proposed by Sha & Lukasiewicz in which they modify sampling layer to use weighted sum of multiple distributions instead of single style vector distribution. This ensure continuity in the latent space.

**Summary Of The Review:**

The paper proposes a simple extension of disentangled variational encoder improving the continuity in latent space. The approach is simple and logical but has little novelty and experiments do not showcase the utility in merging datasets.

---

> ### Author Response · Authors · 2021-11-10
> **Clarification of the expressed doubts.**
>
> Thank you for your review.
>
> Regarding the three key points you highlighted:
> 1. As you mentioned, the mathematics behind our approach is rather simple, but, nonetheless effective and easily understandable. We believe that, compared to GAN and Transformers, our method is able to provide effective disentanglement and explainable embeddings without costly training methods. Furthermore, compared to the model by Sha&Lukasiewicz(2021), we obtain a better representation, also helpful to understand the properties of the latent space. In addition, as previous work that links categorical and continuous annotations through disentanglement is scarce, it is difficult to compare our results with other GAN or Transformers architecture. With our work we hope to establish a useful baseline for future works.
> 2. By disentangling each style type and projecting its value in the same continuous space across multiple datasets it is straightforward to create a single, merged one. Figure 5a shows how the style distributions of the two datasets (respectively blue and red) align perfectly and figure 5b shows, instead, how samples of both datasets are encoded, showing high correlation with the ground truth values. As a consequence, each sample of both datasets can be labelled with the new encoded style vector when creating the new merged one. If you believe this step is not clear enough, we will make it more explicit.
> 3. We employed the same notation used by Sha&Lukasiewicz for better readability and comparison. If you believe there are points in which this notation is not clear enough, we'll provide better explanation.

---

### Decision · Program_Chairs · 2022-01-20

**Decision:**

Reject

**Comment:**

This work proposes a continuous disentanglement variational autoencoder. The approach is a direct extension of Sha & Lukasiewcz (2021). The proposed method appears effective in learning disentangled factors on synthetic data. However, the approach is a minor change to Sha & Lukasiewcz (2021) that samples a weighted sum over all style values. This limits the novelty of the paper. Additionally, evaluation is only on small synthetic datasets that was created for this paper. The lack of evaluation on standard datasets such as an emotion dataset as motivated in the paper, means the results may be due to data selection rather than a superior method. This raises doubts as to whether the approach would generalize to other datasets. In the rebuttal the authors state they wanted to focus on a synthetic dataset since various metrics are easily method but additional real-world/standard dataset results can be added while keeping the synthetic results.